# Seawater Splitting for Hydrogen Generation Using Zirconium and Its Niobium Alloy under Gamma Radiation

**DOI:** 10.3390/molecules27196325

**Published:** 2022-09-26

**Authors:** Imran Ali, Gunel Imanova, Teymur Agayev, Anar Aliyev, Sakin Jabarov, Hassan M. Albishri, Wael Hamad Alshitari, Ahmed M. Hameed, Ahmed Alharbi

**Affiliations:** 1Department of Chemistry, Jamia Millia Islamia (Central University), Jamia Nagar, New Delhi 110025, India; 2Department of Chemistry, King Abdulaziz University, Jeddah 22252, Saudi Arabia; 3Institute of Radiation Problems, Azerbaijan National Academy of Sciences, AZ 1143 Baku, Azerbaijan; 4Department of Chemistry, College of Science, University of Jeddah, P.O. Box 80327, Jeddah 21589, Saudi Arabia; 5Department of Chemistry, Faculty of Applied Sciences, Umm Al-Qura University, Makkah 21955, Saudi Arabia

**Keywords:** seawater splitting, hydrogen generation, Zr1%Nb alloys, γ-radiation, thermal and radiation–thermal decompositions

## Abstract

Hydrogen production is produced for future green energy. The radiation–chemical yield for seawater without a catalyst, with Zr, and with Zr1%Nb (Zr = 99% Nb = 1%) were (G(H_2_) = 0.81, 307.1, and 437.4 molecules/100 eV, respectively. The radiation–thermal water decomposition increased in γ-radiation of the Zr1%Nb + SW system with increasing temperature. At T = 1273 K, it prevails over radiation processes. During the radiation and heat radiation heterogeneous procedures in the Zr1% Nb + SW system, the production of surface energetic sites and secondary electrons accelerated the accumulation of molecular hydrogen and Zr1%Nb oxidation. Thermal radiation and thermal processes caused the metal phase to collect thermal surface energetic sites for water breakdown and Zr 1%Nb oxidation starting at T = 573 K.

## 1. Introduction

Due to the increased and pressing demand for a clean environment, there is a great need for green energy. Additionally, the estimate of the exhaustion of fossil fuels in the future is compelling researchers to explore economic green energy. The global energy requirement for sustainable development, i.e., clean and affordable energy, is estimated to double by 2050. Therefore, the exploration of clean and renewable energy is an urgent task. Amongst many forms of energy, hydrogen is considered the best due to its high energy content per unit of weight. Researchers and governments of different countries are making it a top priority. For hydrogen production, there is a need for raw materials, and seawater is freely available in abundance. In most studies, water-splitting researchers use catalysts to decompose water under the influence of radiation. This is the era of nanomaterials being used in many applications [1,2,3,4]. Many materials have been utilized for the photodecomposition of water with various types of radiation [5,6,7]. The most important radiations used are UV, visible, IR, X-ray, gamma etc. In early works [8,9,10,11], the results of thermal studies of the properties of zirconium and zirconium alloy samples were carried out in the temperature range of T= 600–1200 °C. Zirconium and its alloys have several features in corrosion processes that are not observed in other metals and alloys. Despite the numerous publications on the issue of the investigation of corrosion resistance of zirconium and zirconium alloys, the study of this unique material continues. Zirconium and its alloys are the main structural materials used in the nuclear industry for the production of fuel (fuel rods), elements of nuclear reactors. Zirconium is a fairly active metal that reacts with oxygen, nitrogen, water vapor, carbon dioxide, hydrogen, and hydrocarbons [12,13,14]. However, using zirconium and zirconium alloy as structural materials for reactors promotes the formation of durable and chemically resistant oxide films on the metal surface, protecting them from corrosion promoters. Krasnoruyky and Petelguzov [15] reported the effects of corrosion rate and thickness of oxide films under similar conditions on changes in the structure of zirconium and zirconium alloy after heating up to a certain temperature. The term oxidation was first used to describe reactions in which metals react with oxygen in the air to yield metal oxides. Iron rusts; aluminum develops a layer in air, and the metal gains a positive charge by transporting electrons to oxygen atoms. This results from a negative charge on oxygen (O^2−^). Consequently, meat and oxygen oxidized and reduced, respectively [16]. Al-Azmi and Sajjad [17] used gold-deposited S-doped graphene quantum dots for water-splitting to obtain hydrogen. The authors reported a good yield of hydrogen. Keshipour and Asghari [18] used nickel (II) phthalocyanine (NiPc) graphene oxide (GO)/TiO_2_. As per the authors, GO reduced the band gap of TiO_2_ by donating to Fermi levels, and NiPc escalated the photocatalytic process as a sensitizing agent [19].

Among many materials, zirconium and its alloys are gaining importance in many applications. Zirconium combinations are broadly utilized as basic materials for central items of the dynamic zone, water-cooled atomic control plants, high temperatures, and heat and mechanical loads of the coolant and other favorable components. In recent decades, numerous multilayer coatings have been systematically examined concerning radiation-induced flaws and the accompanying changes in mechanical characteristics in irradiated materials [20,21]. These multi-crystal structures, in which vacancies and interstitials can recombine, are frequently employed as self-healing coverings in radiation point defects [6]. On the basis of this idea, metals with different crystal structures were mixed to create nanoscale multilayer coatings with greater tolerance to radiation damage brought on by helium ion and proton bombardment [22,23,24]. Keeping these facts into consideration, we examined the radiation decomposition of seawater with and without Zr and Zr1%Nb catalysts at distinctive temperatures under γ-quanta.

## 2. Materials and Methods

The experiments were performed in inactive situations in specialized quartz vessels with V = 1.0 cm^3^ [25,26]. The metallic zirconium and Zr1%Nb combinations were taken as the catalysts. The catalysts were characterized before the experiments. Additionally, these catalysts were also characterized after the experiments to observe the changes in the structures. The X-ray experiments were performed in an X-Ray Diffractometer D2 PHASER (model number Bruker D8 Advance; Bruker, Billerica, MA, USA.). The sample chips were set within the goniometer of the diffractometer. The X-ray diffraction range of the test was drawn within the extent of the diffraction point 20 < 2θ < 100. The cage parameters were calculated based on the square equations of the crystallography. Then, based on the obtained X-ray diffraction spectra, the different parameters of the sample were determined. Of these, the distances between the atomic flatness (d) and intensities were determined. Thus, syngonia—to which the sample belongs—the lattice size, density, lattice constants, and the spatial group were determined from XRD spectra (Bruker, Billerica, MA, USA).

The weight of zirconium and Zr1%Nb combinations were in the array of 0.0798–0.0825 g. The contact surfaces of the catalysts were decided on the premise of geometric measurements, and these were 8.47 cm^2^/g and 9.05 cm^2^/g for Zr1%Nb. The quartz ampoules were cleaned with ethanol and acetone and solvents followed by deionized water washing. The quartz ampoules were dried with inert gas at T = 300–320 K. The samples were placed in quartz vessels filled with seawater and vacuum-treated at T = 373 and 673 K at 10^−2^ Pascal pressure. The ampoules were filled with 5.5 × 10^−2^ g of water. Seawater inoculation into the vessels was accurate to within 2%. Until solvable oxygen and other vapors (organic substances) were eliminated from the water via the freeze–pump–thaw series, the ampoules containing the samples were delimited. With a precision of 10 °C, the temperature was maintained during the experiments. Using a ^60^Co isotope source, radiation and thermal radiation procedures were carried out. Chemical dosimeters were used to assess the radiation source’s absorbed dosage intensity.

Column [25].: C.-1010 P., 30 m × 0.53 mm I.D. (25467), D.—TCD, C. Flow, 3 mL/min, 20 Hz/0.01 min, T (head) = 500 °C, T(d.) = 2300 C, M.F. = 0 mL/min, gas car.—Ar, oven: 500 °C (7.0 min), 200 °C/min to 2300 °C, i. temp.: 2300 °C. 

## 3. Results and Discussion

### 3.1. Characterization of the Catalysts

The structure determination of Zr metal and Zr-1%Nb alloy was carried out before and after the experiments. The results of XRD analysis of Zr metal and Zr-1%Nb alloy are shown in Figure 1a,b, respectively, at room temperature (T = 300 K) and under normal conditions. It is clear from Figure 1a that the XRD pattern had diffraction peaks at 2θ values of 24.10°, 29.4°, 31.0°, 34.1°, 35°, 38°, 40°, 45°, 45.5°, 49°, 50°, 54°, 55°, 57°, 58°, and 60°, corresponding to the monoclinic symmetry P21/A, A = 5.31 Å; B = 5.21 Å; C = 5.15 Å; β(beta) = 99.23, and Z = 4-phase groups, respectively. Figure 1b shows the X-ray diffraction spectrum of the Zr1%Nb alloy at room temperature (T = 300 K) and in normal conditions. The XRD pattern had diffraction peaks at 2θ values of 30.5°, 35.1°, 36.0°, 51.0° and 60° corresponding to the cubic symmetry P42/nmc, A = 3.6 Å; C = 5.18 Å; V = 67.33; Z = 2 phase groups. The difference between the structures of Zr metal and Zr-1%Nb alloy was observed, as were the symmetry of Zr metal and the cubic symmetry of Zr-1%Nb alloy. However, both catalysts have enough surface area, which is responsible for strong catalytic activities. These results are in agreement with earlier research [27]. A similar characterization of Zr metal and Zr-1%Nb alloy (catalysts) was carried out after the hydrogen generation experiments. It was observed that there was no change in the peak intensities or positions, confirming no change in the structures of the catalysts after the experiments.

### 3.2. Hydrogen Generation

The hydrogen generation from seawater was studied under the influence of gamma radiation. The decomposition was carried out without any catalyst, with pure zirconium, and with Zr 1%Nb alloy. The results are discussed in the following paragraphs.

#### 3.2.1. Seawater Decomposition without Any Catalyst

The radiolysis of the saltwater was first performed without the use of a catalyst. Figure 2 depicts the dynamic curve of atomic hydrogen accumulation during radiolysis of pure seawater. Based on the linear kinetic section, the hydrogen radiation–chemical yield and its molecular hydrogen output were calculated. The corresponding numbers were 5.30 × 10^13^ molecules/g and 0.81 molecules/100 eV, respectively. These results were contrasted with those obtained under the same circumstances using pure water. Under the same circumstances, pure water produced radiation–chemical yields of 0.41 to 0.45 molecules/100 eV. The contribution of salts dissolving in water can be used to explain the observed increase in molecular hydrogen output [28]. Since neither carbs nor CO nor CO_2_ were the observed products, organic contaminants had no bearing. The consequence of ion–dipole interactions among inorganic salts and water can be used to explain the involvement of Na+, K+, Mg+, and Ca+ cations in seawater. The density of electrons of water molecules is shifted near the acceptor levels of captions via interactions between captions and water dipoles. Due to this interaction, the energy characteristics of water and the decomposition processes (bond energy, ionization potential) changed. As a result, radiolysis of seawater produced a larger radiation–chemical yield of molecular hydrogen than radiolysis of our water.

#### 3.2.2. Seawater Decomposition in Presence of Zr 

Furthermore, attempts were made to increase hydrogen yield by using catalysts. As mentioned above, Zr was used as the catalyst, and the experiments were carried out under varying temperatures ranging from T = 473 to 1273 K. The experiments were carried out under thermal, radiation, and both radiation and thermal conditions. The results are summarized in Table 1. The values of the velocity of the thermal process [W_T_(H_2_)] ranged from 4.25 × 10^14^ to 5.5 × 10^17^ molecules/g.s at 300 to 1273 K. The values of the velocity of the radiation process cess [W_R_(H_2_)], ranged from 5.19 × 10^13^ to 2.1 × 10^17^ molecules/g.s at T = 300 to 1273 K. The values of the velocity of the radiation–thermal process; [W_RT_(H_2_)] ranged from 9.44 × 10^14^ to 5.45 × 10^17^ at molecules/g.s T = 473 to 1273 K. It was observed that the values of W_RT_(H_2_) increased with increasing temperature. The rate of decomposition of water was higher in the case of the radiation phenomenon in comparison to the thermal process. However, the rate of seawater decomposition increased many times when both thermal and radiation processes were carried out simultaneously. This rise was caused by the formation of positively charged ions in the catalyst under the impact of radiation as a result of the adsorption of water and the transfer of charges to water molecules. The recombination of these positively charged ions with the generated surface electrons disintegrates the water molecules. The cost of the electrons and the charges created on the surface in this process influence how much hydrogen can be produced. The mobility of the particles on the catalyst surfaces increases with temperature. The values of radiation–chemical yields [G(H_2_)] at T = 473 to 1273 K ranged from 7.90 to 307.1 molecules/100 eV.

#### 3.2.3. Seawater Decomposition in the Presence of Zr1%Nb Alloy

Furthermore, efforts were made to increase hydrogen yield, and for this purpose, a zirconium alloy was used. Seawater decomposition in the presence of Zr1%Nb was performed to determine the role of radiation-heterogeneous procedures in seawater decomposition. The experiments were carried out under varying temperatures ranging from T = 473 to 1273 K. The experiments were performed in thermal, radiation, and radiation–thermal processes. The strength of the sorbet radiation dose was determined for the total Zr1%Nb alloys + SW system with a permanence for the electron density of its constituents. 

The mass unit of the overall system Ni(H_2_) = NH_2_/mH_2_O + m_Zr_ and the kinetic graphs of accumulation Ni(H_2_) = f were used to compute the amount of hydrogen molecules that were obtained (t). Thermal breakdown of water was seen to take place beneath the Zr1% Nb alloys + SW contact as the process temperature increased (above 473 K). As a result, the Zr1%Nb alloys + SW system underwent identical thermal and heat radiation to hydrogen generation experiments at temperatures exceeding 473 K. Figure 3a–c, respectively, depict the kinetic graphs of molecular hydrogen buildup in thermal (Figure 3a), radiation (Figure 3b), and radiation–thermal processes (Figure 3c) of disintegration of water in the Zr1%Nb alloys + SW system. The rates and chemical yields of hydrogen radiation were calculated using kinetic data.

For the difference between thermal radiation and thermal processes rates of molecular hydrogen buildup, the rates of radiation constituents in radiation–thermal processes were determined:W_R_(H_2_) = W_RT_(H_2_) − W_T_(H_2_)

In Table 2, a list of the kinetic parameters for thermal, radiation, and radiation–thermal processes is provided. Temperature changes led to higher radiation–chemical yields and faster rates of hydrogen buildup. When in contact with Zr or Zr1%Nb alloys, the rates of heat reactions and water degradation also accelerated. After a comparison and calculation of Zr and Zr1%Nb alloy, it was found that the effect of temperature and radiation on the seawater with Zr1%Nb alloy is very strong, which enhances the radiolysis process.

At T = 473 K, water breakdown caused by radiation is the main mechanism. The following is a schematic representation of the radiolysis mechanisms involved in the disintegration of water at these temperatures:

Seawater radiolysis under γ radiation:H_2_O_s_ → H_2_ + ∑Π_i,_(1)
W_R_(H_2_) = G(H_2_)D × 10^−2^
where H_2_O is seawater, ∑Π_i_ is radiolysis seawater products, G(H_2_) is the hydrogen yield of radiation–chemical, and D is the strength of the sorbet dose of γ radiation.

Active–active states surface creation of and secondary electron release in γ-radiation:HC → S^*^ + e_sec._(2)
where S^*^ = surface-active state and e_sec._ = secondary electron release from a metal phase in irradiation.

Water molecules decomposition owing to interactions with active states and e_sec._ produced due to:H_2_O_s_ + S^*^ → H_2_ + ∑Π_i_ + S(3)
where S is the relaxed state of the surface-active centers:H_2_O_s_ + e_sec_ → H_2_ + ∑Π(4)
W_3_(H_2_) = k_3_[H_2_O_s_][S^*^] + k_3_[H_2_O_s_][e_sec_]

Active sites are also collected on Zr1%Nb alloys surface at T ≥ 573 K:Zr1%Nb → S_T_^*^(5)
S_T_^*^ + H_2_O_s_ → H_2_ + ∑Π + S(6)
W_T_ = k_T_[S^*^_T_][H_2_O_s_]
where S_T_^*^ = thermally stimulated state of Zr1%Nb surface and S = relaxed state of thermally stimulated states.

The activation energy barrier for thermal processes that generate active site surfaces normally exists and increases with temperature. The equation for the hydrogen gathering rate can be stated as follows when the aforementioned phases of the radiation and thermal constituents of the procedures are taken into account:W(H_2_) = G(H_2_)D × 10^−2^ + [H_2_O](k_3_[S^*^_T_] + k_4_[e_sec_])
W_T_(H_2_) = k_T_[S_T_^*^[H_2_O]]

The affiliation between the rates of the radiation and thermal constituents is:W_T_(H_2_)/W_RT_(H_2_) = G(H_2_)D × 10^−2^/ k_T_[S_T_^*^[H_2_O]] + k_3_[S_T_^*^] + k_T_[S_T_^*^] + k_4_[e_sec_]/k_T_[S_T_^*^]

The ratios of rates of radiation and thermal processes depend on the density of active sites on the metal’s surface and the strength of secondary electron release from it because many parameters in these expressions are persistent. The radiation processes’ diffusion stages are stimulated by rising temperatures. The temperature range at which imperfection states travel in the crystal lattice at T = 673 K is where the temperature effect on radiation constituents of the procedures is most likely to occur. 

The results indicate that at T = 1273 K, radiation–thermal processes dominate the radiation process. As a result, it is impossible to calculate at the rates of the radiation constituents or radiation–chemical yield of hydrogen at T = 1273 K. The activation energies were determined by rates of procedures in the Arrhenius coordinates as a purpose of temperature.

## 4. Activation Energy of Water Decomposition

The rates of (1) thermal and (2) radiation–thermal processes of molecular hydrogen accumulation during the radiation-heterogeneous breakdown of seawater in the presence of Zr1%Nb are dependent upon one another, as shown by the activation energy of seawater decomposition (Figure 4). The radiation–thermal and thermal processes that led to molecular hydrogen buildup had activation energies of Ea = 10.7 and 38.5 kJ/mol, respectively. Two zones can be found on the temperature graph for the radiation processes rate. The first one it at T = 300–573 K with Ea = 10.7 kJ/mol, and the second one is at T = 773–1273 K with Ea = 38.5 kJ/mol. Thermal decay of seawater in interaction with Zr1% Nb had larger activation energy than radiation–thermal and radiation processes did.

## 5. Kinetics of Water Decomposition

The breakdown of seawater by radiation was aided by secondary electron emissions and radiation–thermal active sites on the surface, which have larger kinetic energies than thermally active sites. The kinetics of Zr1% Nb corrosion at T = 673 K was investigated gravimetrically in the Zr1%Nb + SW system to clarify the consistencies of surface processes. The kinetic curves for Zr1%Nb alloys under thermal and radiation–thermal processes are shown in Figure 5. These curves (W_T_ = 2.4 × 10^−6^ g/sm^2^*s and W_RT_ = 4.5 × 10^−4^ g/sm^2^*s, respectively) were used to calculate the rates of Zr1%Nb corrosion in thermal and radiation–thermal procedures in commerce with seawater.

The amount of molecular hydrogen and the alterations in sample mass following thermal, radiation, and radiation–thermal processes in commerce with SW can both be used to describe Zr1%Nb corrosion. The following formula [25] describes the kinetic curves indicating alterations in the sample mass in radiation–thermal processes:B(t) = B_∞_(1 − exp(−k_c_t))
where B_∞_ is the amount of product. In these experiments, that may be either oxide film or molecular hydrogen at t → ∞, k_c_− = corrosion rate constant, k_c_ = k_0_e ^−E^_g_^/RT^ [4], k_0_ = 10^−2^s^−1^, and k_B_ = Boltzmann constant.

The kinetic curves’ beginning portions have a linear dependency: B(t) = f(t). The steady-state zone in dependences Bi = f is detected in our trials at 120 min. Consequently, the following inferences can be concluded:

Compared to pure water, soluble salts in seawater boosted the radiation–chemical production of molecular hydrogen.

The radiation–chemical output of hydrogen and metal corrosion was raised by radiation-heterogeneous processes in Zr1%Nb with seawater.

In addition to the radiation activities, thermal processes for the creation of surface-active states also took place at T = 523 K as the temperature of radiation-heterogeneous processes increased.

As the temperature of the Zr1%Nb + SW systems rose, the contribution of thermal processes increased until, at T = 1273 K, it predominated over the radiation contribution.

## 6. Mechanism

The results can be explained using the radiation chemistry’s known mechanisms. Depending on the scattering angle, Compton electron energies range from 0 to 1.02 eV. The Compton electrons gradually lose their kinetic energies in both elastic and inelastic collisions and transform into radiation electrons in the Zr1%Nb + seawater system depending on their kinetic energy, passing from the liquid phase into the seawater multiple times or vice versa. Consequently, sorption of water and charge transmission to water molecules results in radiation and positively charged ions being created in the catalyst. The recombination of these positively charged ions with the generated surface electrons results in the disintegration of the water molecules at this stage. As the temperature increases, the movement of the particles on the catalysts surface increases. Contrarily, as part of the electron–hole pair elaborated in the water decomposition is renewed, these tempted particles (by radiation) contributed to water decomposition. Therefore, hydrogen creation at higher temperatures was superior. The literature supports many publications on water decomposition for hydrogen production, but there is still no technique that can produce hydrogen at a large scale [23]. It is important to mention that the metal and its alloy oxidized during the process.

## 7. Conclusions

Many world scientists have conducted other studies of the sample we used, and they report on the first fabrication of vertically oriented niobium–zirconium oxynitride nanotube arrays and their use as an attractive and robust material for visible-light-driven water oxidation [29]. We studied other physical and chemical properties of that sample and showed the results we obtained in the article we presented. The contributions of thermal and radiation–thermal processes during the gathering of molecular hydrogen and corrosion of alloys Zr1%Nb in contract with seawater were discovered. It has been established that under gamma irradiation of a system of Zr1% Nb alloy with seawater, with increasing temperature, the involvement of thermal processes of accumulation of molecular hydrogen also increased at T ˃ 1273 K, which prevails over radiation–thermal processes. Nevertheless, the rate of the gathering of thermally active sites increased with temperature, and the results showed that the thermal procedures dominated over the radiation–thermal procedure W_T_ ≥ W_RT_ at T = 1273 K. Henceforth, the rates of radiation components of radiation–chemical yield of hydrogen can be established at T ≥ 1273 K. Thus, the Zr1%Nb + seawater system can be considered relevant for hydrogen production in the future.

## Figures and Tables

**Figure 1 molecules-27-06325-f001:**
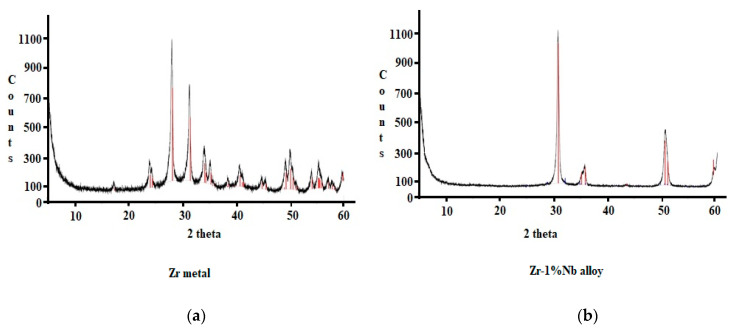
X-ray diffraction spectra of (**a**) Zr metal and (**b**) Zr-1%Nb alloy at room temperature.

**Figure 2 molecules-27-06325-f002:**
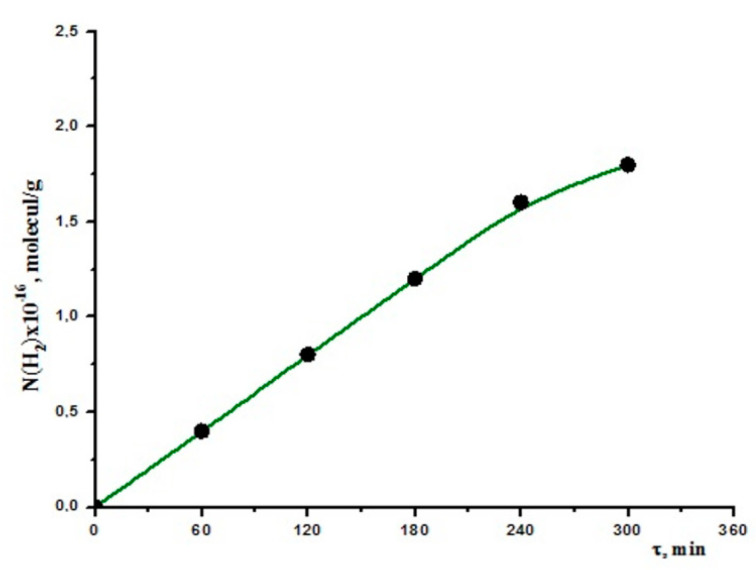
Kinetics of molecular hydrogen of seawater decomposition without catalyst at T = 300 K, D = 1.05 Gy/s.

**Figure 3 molecules-27-06325-f003:**
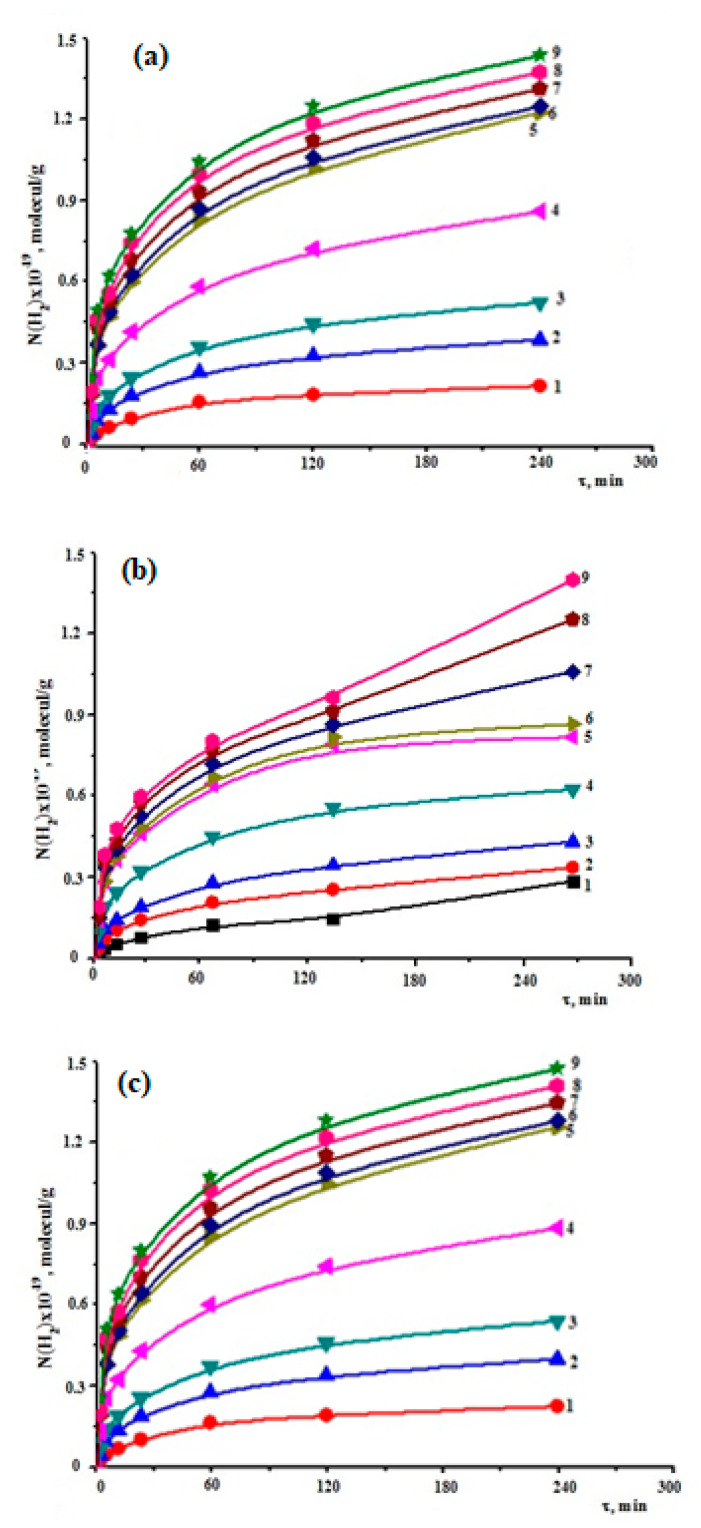
Kinetics of accumulation of molecular hydrogen during (**a**) thermal, (**b**) radiation, and (**c**) radiation–thermal decomposition of seawater on the surface of Zr1%Nb alloys at different temperatures: 1: 473 K; 2: 573 K; 3: 673 K; 4: 773 K; 5: 873 K; 6: 973 K; 7: 1073 K; 8: 1173 K; and 9: 1273 K.

**Figure 4 molecules-27-06325-f004:**
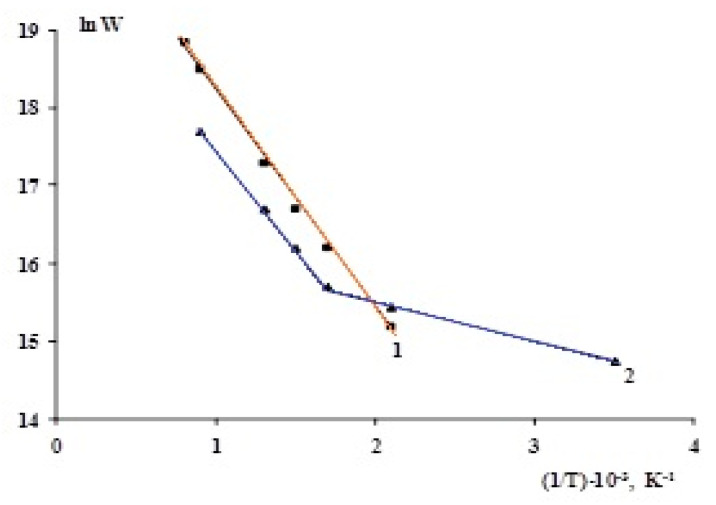
Temperature dependences of molecular hydrogen production during thermal (**1**) and radiation–thermal (**2**) processes in the Zr1%Nb + seawater system.

**Figure 5 molecules-27-06325-f005:**
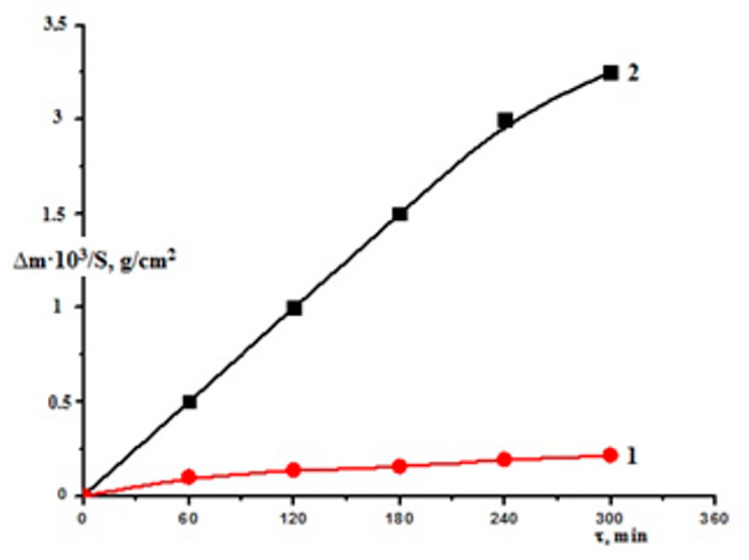
Kinetic curves of Zr1%Nb in (**1**) thermal and (**2**) radiation–thermal processes in seawater.

**Table 1 molecules-27-06325-t001:** The results of thermal, radiation, and radiation–thermal procedures and radiation–chemical yields of hydrogen in seawater decomposition with Zr.

T (K)	W_T_(H_2_)	W_R_(H_2_)	W_RT_(H_2_)	G(H_2_), molec./100 eV
**473**	4.25 × 10^14^	5.19 × 10^14^	9.44 × 10^14^	7.90
**573**	1.1 × 10^15^	1.08 × 10^15^	2.18 × 10^15^	16.1
**673**	2.3 × 10^15^	2.7 × 10^15^	5.0 × 10^15^	39.5
**773**	5.8 × 10^16^	4.2 × 10^16^	1.02 × 10^17^	60.8
**873**	1.2 × 10^1^^7^	7.0 × 10^16^	1.9 × 10^17^	101.4
**973**	1.8 × 10^1^^7^	6.0 × 10^16^	3.0 × 10^17^	106.7
**1073**	2.0 × 10^17^	0.9 × 10^17^	4.1 × 10^17^	304.3
**1173**	3.0 × 10^17^	1.1 × 10^17^	4.4 × 10^17^	306.6
**1273**	5.5 × 10^17^	2.1 × 10^17^	5.45 × 10^17^	307.1

W_RT_(H_2_) = the velocity of the radiation–thermal process; W_T_(H_2_) = the velocity of the thermal process; W_R_(H_2_) = the velocity of the radiation process; and G(H_2_) = radiation–chemical yields.

**Table 2 molecules-27-06325-t002:** The results of thermal, radiation, and radiation–thermal procedures and radiation–chemical yields of hydrogen in seawater decomposition with Zr1%Nb alloy.

T (K)	W_T_(H_2_)	W_R_(H_2_)	W_RT_(H_2_)	G(H_2_), molec./100 eV
**473**	5.2 × 10^14^	5.90 × 10^14^	11.1 × 10^14^	9.00
**573**	1.6 × 10^15^	1.3 × 10^15^	2.91 × 10^15^	19.7
**673**	2.8 × 10^15^	3.3 × 10^15^	6.1 × 10^15^	48.1
**773**	6.5 × 10^16^	6.3 × 10^16^	1.28 × 10^17^	91.3
**873**	1.35 × 10^17^	1.25 × 10^17^	2.6 × 10^17^	181.1
**973**	1.49 × 10^17^	1.3 × 10^17^	3.3 × 10^17^	185.1
**1073**	3.5 × 10^17^	3.0 × 10^17^	6.5 × 10^17^	434.7
**1173**	4.5 × 10^17^	3.1 × 10^17^	6.7 × 10^17^	434.9
**1273**	6.83 × 10^17^	3.2 × 10^17^	6.9 × 10^17^	437.4

W_RT_(H_2_) = the velocity of the radiation–thermal process; W_T_(H_2_) = the velocity of the thermal process; W_R_(H_2_) = the velocity of the radiation process and G(H_2_) = radiation–chemical yields.

## Data Availability

Not applicable.

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
