# Peer review of "Seawater Splitting for Hydrogen Generation Using Zirconium and Its Niobium Alloy under Gamma Radiation"

_molecules, 2022, doi:10.3390/molecules27196325_

Round 1
Reviewer 1 Report
In general, in the paper, the authors comments the preparation of the materials to use in the process to produce H2 from seawater. In their results the authors shows some interesting results in the line with the global environmental objective. However, the authors do not have any comments from the the physical or chemical properties of the materials.
Crystallinity properties of the materials?, DRX characterization?
How are the metals in the surface of the alloy? oxidized or reduced?. H2-RTP?
Real percentages of the metals in the alloy-. Elemental analysis
Also, the selection of these materials are not very well supported in the introduction section. The introduction has very short and very general information without any details.
Author Response
In general, in the paper, the authors comments the preparation of the materials to use in the process to produce H2 from seawater. In their results the authors shows some interesting results in the line with the global environmental objective. However, the authors do not have any comments from the the physical or chemical properties of the materials.
Reply: Thanks to this reviewer for his/her appreciation of our work.
Also, thanks for sparing his/her valuable time reviewing this manuscript and giving fruitfulsuggestions.
Crystallinity properties of the materials?, DRX characterization?
Reply:The materials (zirconium and its alloy) were purchased and used as such. Therefore, it is out of scope of the article.
How are the metals in the surface of the alloy? oxidized or reduced?. H2-RTP?
Reply:The metal and its alloy oxidized during the process.
Real percentages of the metals in the alloy-. Elemental analysis
Reply:Zr1%Nb (Zr = 99% Nb = 1%)
Also, the selection of these materials are not very well supported in the introduction section. The introduction has very short and very general information without any details
Reply:Corrected in the article
Once Again We are highly thankful to this reviewer for sparing HIS/H

Reviewer 2 Report
I think an interesting study was done on the seawater conversion to hydrogen by Zr/Nb catalyst. The seawater could be a promising source for this valuable reaction. So, I believe this work could be worthy for the Molecule's readers after corrections as below:
1. There are a lot of English mistakes, please correct them. For example, just for the abstract the following issues were detected:
a) The first paragraph of the abstract needs revision since the hydrogen generation is not a dream and that is done now.
b) page 1, line 10, “the future” is correct.
c) Page 1, line 10, “Amongst” instead of “among”
d) Page 1, line 14, “100eV, respectively.” is correct.
f) Please add a space before all units like “100 eV” not “100eV”.
2. The introduction is very brief and the number of references is very low. Please more discuss about the clean hydrogen generation and various aspect of hydrogen generation by radiation. The following references are some recent references around green hydrogen generation, please add them.
a) Journal of Colloid and Interface Science 2022, 612, 701-709
b) International Journal of Hydrogen Energy 2022, 47, 12865
c) Scientific Reports 2021, 11, 16148
3. Please add the material and method section.
4. Why you used 1% of Nb? The catalyst optimization should be added. You may obtain better results with higher amounts of Nb.
5. The mechanism needs a reference.
6. The results should be compared with the previous reports to more highlight the importance of the work. In particular, the results should be compared with the similar catalysts like the following reference.
https://doi.org/10.1021/acsanm.0c01282
Author Response
Pointwise replies
Manuscript ID: molecules- 1868835
Title: Seawater splitting for hydrogen generation using zirconium and its
niobium alloy under gamma radiation
First of all, I would like to thank Ms. Astrid Zhao, Assistant Editor, to give us a chance for revising this manuscript. Besides, thanks are also the scholarly reviewer to give fruitful suggestions. Really, the incorporation of all the suggestions made this manuscript more useful and attractive to the readers. The point-wise replies to the comments of reviewers are given below.
Open Review
( ) I would not like to sign my review report
(x) I would like to sign my review report
English language and style
(x) Extensive editing of English language and style required
( ) Moderate English changes required
( ) English language and style are fine/minor spell check required
( ) I don't feel qualified to judge about the English language and style
Yes |
Can be improved |
Must be improved |
Not applicable |
|
Does the introduction provide sufficient background and include all relevant references? |
( ) |
( ) |
(x) |
( ) |
Are all the cited references relevant to the research? |
(x) |
( ) |
( ) |
( ) |
Is the research design appropriate? |
( ) |
(x) |
( ) |
( ) |
Are the methods adequately described? |
( ) |
(x) |
( ) |
( ) |
Are the results clearly presented? |
(x) |
( ) |
( ) |
( ) |
Are the conclusions supported by the results? |
(x) |
( ) |
( ) |
( ) |
Comments and Suggestions for Authors
I think an interesting study was done on the seawater conversion to hydrogen by Zr/Nb catalyst. The seawater could be a promising source for this valuable reaction. So, I believe this work could be worthy for the Molecule's readers after corrections as below:
Reply:
Thanks to this reviewer for his/her appreciation of our work.
Also, thanks for sparing his/her valuable time reviewing this manuscript and giving fruitful suggestions.
- There are a lot of English mistakes, please correct them. For example, just for the abstract the following issues were detected:
Reply:
Englished is improved.
- The first paragraph of the abstract needs revision since the hydrogen generation is not a dream and that is done now.
Reply:
Corrected in the article
- b)page 1, line 10, “the future” is correct.
Reply:
Yes
- c)Page 1, line 10, “Amongst” instead of “among”
Reply:
Corrected in the article
- d)Page 1, line 14, “100eV, respectively.” is correct.
Reply:
Yes
f) Please add a space before all units like “100 eV” not “100eV”.
Reply:
Corrected in the article
- The introduction is very brief and the number of references is very low. Please more discuss about the clean hydrogen generation and various aspect of hydrogen generation by radiation. The following references are some recent references around green hydrogen generation, please add them.
- a)Journal of Colloid and Interface Science 2022, 612, 701-709
Reply:
Added in the revised manuscript.
- b)International Journal of Hydrogen Energy 2022, 47, 12865
Reply:
Added in the revised manuscript.
- Scientific Reports 2021, 11, 16148
Reply:
Added in the revised manuscript.
- Please add the material and method section.
Reply:
Added in the revised manuscript.
- Why you used 1% of Nb? The catalyst optimization should be added. You may obtain better results with higher amounts of Nb.
Reply:
The alloy was supplied by the company and it was only available i.e. 1%Nb.
- The mechanism needs a reference.
Reply:
Added in the revised manuscript.
- The results should be compared with the previous reports to more highlight the importance of the work. In particular, the results should be compared with the similar catalysts like the following reference.
https://doi.org/10.1021/acsanm.0c01282
Reply:
Added in the revised manuscript.

Round 2
Reviewer 1 Report
The authors have included some information in the introduction section, however, the materials characterization section has not been improved. The authors have only provided general information about the material but without scientific support. I consider that the current format is not sufficient to be published in molecules
Author Response
The authors have included some information in the introduction section, however, the materials characterization section has not been improved. The authors have only provided general information about the material but without scientific support. I consider that the current format is not sufficient to be published in molecules
Reply:
Thanks to this reviewer for his/her appreciation of our work.
Also, thanks for sparing his/her valuable time reviewing this manuscript and giving fruitful suggestions.
We have added some more information in the introduction. Also, we included a new section on characterization. The structures of the catalysts were determined before and the experiments.
Reviewer 2 Report
The manuscript is acceptable.
Author Response
Pointwise replies
Manuscript ID: molecules- 1868835
Title: Seawater splitting for hydrogen generation using zirconium and its
niobium alloy under gamma radiation
First of all, I would like to thank Ms. Astrid Zhao, Assistant Editor, to give us a chance for revising this manuscript. Besides, thanks are also the scholarly reviewer to give fruitful suggestions. Really, the incorporation of all the suggestions made this manuscript more useful and attractive to the readers. The point-wise replies to the comments of reviewers are given below.
Open Review
( ) I would not like to sign my review report
(x) I would like to sign my review report
English language and style
( ) Extensive editing of English language and style required
( ) Moderate English changes required
(x) English language and style are fine/minor spell check required
( ) I don't feel qualified to judge about the English language and style
Yes |
Can be improved |
Must be improved |
Not applicable |
|
Does the introduction provide sufficient background and include all relevant references? |
(x) |
( ) |
( ) |
( ) |
Are all the cited references relevant to the research? |
(x) |
( ) |
( ) |
( ) |
Is the research design appropriate? |
(x) |
( ) |
( ) |
( ) |
Are the methods adequately described? |
(x) |
( ) |
( ) |
( ) |
Are the results clearly presented? |
(x) |
( ) |
( ) |
( ) |
Are the conclusions supported by the results? |
(x) |
( ) |
( ) |
( ) |
Comments and Suggestions for Authors
The manuscript is acceptable.
Submission Date
31 July 2022
Date of this review
17 Aug 2022 12:51:06
Reply:
Thanks to this reviewer for his/her appreciation of our work.
Also, thanks for sparing his/her valuable time reviewing this manuscript and giving fruitful suggestions.
We are grateful to accept our manuscript.

Round 3
Reviewer 1 Report
Accept in present form
Thank you for the attention and achieving a better work